# Massive and rapid COVID-19 testing is feasible by extraction-free SARS-CoV-2 RT-PCR

Ioanna Smyrlaki[1,5], Martin Ekman[2,5], Antonio Lentini [1], Nuno Rufino de Sousa [3], Natali Papanicolaou[1], Martin Vondracek[2], Johan Aarum[2], Hamzah Safari[2], Shaman Muradrasoli[4], Antonio Gigliotti Rothfuchs [3], Jan Albert [2,3], Björn Högberg [1] & Björn Reinius [1✉]

Coronavirus disease 2019 (COVID-19), caused by severe acute respiratory syndrome coronavirus 2 (SARS-CoV-2), is commonly diagnosed by reverse transcription polymerase chain reaction (RT-PCR) to detect viral RNA in patient samples, but RNA extraction constitutes a major bottleneck in current testing. Methodological simplification could increase diagnostic availability and efficiency, benefitting patient care and infection control. Here, we describe methods circumventing RNA extraction in COVID-19 testing by performing RT-PCR directly on heat-inactivated or lysed samples. Our data, including benchmarking using 597 clinical patient samples and a standardised diagnostic system, demonstrate that direct RT-PCR is viable option to extraction-based tests. Using controlled amounts of active SARS-CoV-2, we confirm effectiveness of heat inactivation by plaque assay and evaluate various generic buffers as transport medium for direct RT-PCR. Significant savings in time and cost are achieved through RNA-extraction-free protocols that are directly compatible with established PCR-based testing pipelines. This could aid expansion of COVID-19 testing.

---

[1] Department of Medical Biochemistry and Biophysics, Karolinska Institutet, 171 77 Stockholm, Sweden. [2] Department of Clinical Microbiology, Karolinska University Hospital, 171 76 Stockholm, Sweden. [3] Department of Microbiology, Tumor and Cell Biology, Karolinska Institutet, 171 77 Stockholm, Sweden. [4] Public Health Agency of Sweden, 171 82 Solna, Sweden. [5]These authors contributed equally: Ioanna Smyrlaki and Martin Ekman. ✉email: bjorn.reinius@ki.se

The emergence of the novel human coronavirus in late 2019 in the Wuhan region of China rapidly evolved into a global pandemic. The high transmission rate and high proportion of asymptomatic infections led to a massive, worldwide need for rapid, affordable, and efficient diagnostic tests, that can be performed in clinical and non-clinical settings[1,2].

Currently, the widely used method of SARS-CoV-2 detection in clinical diagnostics is an RT-PCR assay, detecting the presence of viral RNA in patient samples. Although RT-PCR is widely implemented for the detection of pathogens, including viruses[3] in clinical samples, the implementation of the specific assay for the detection of SARS-CoV-2 has only recently been established. The most commonly used protocol[4] was developed and optimized for the detection of the novel coronavirus at the Charité University Hospital, in collaboration with institutes in Germany, the Netherlands, China, France, the United Kingdom, and Belgium. A different test protocol was developed by the Center for Disease Control (CDC) in the United States through comparison and validation of various kits for nucleic acid extraction and the use of alternative probe and primer sets for SARS-CoV-2 detection in clinical samples[5,6]. Routinely, the application of quantitative PCR (qPCR) for the relative quantification of an RNA of interest is preceded by (1) the isolation and purification of total RNA from the sample, (2) elution and possible concentration of the material, and (3) the use of purified RNA in a reverse-transcription (RT) reaction resulting in complementary DNA (cDNA) from the template RNA which is then utilized for the qPCR reaction. However, nucleic acid purification and RT of the resulting RNA into cDNA are not only laborious and time-consuming, but the additional steps requiring manual handling can result in experimental errors. In the case of clinical sampling and diagnostics, the use of a single-reaction kit combining the RT and qPCR reactions is therefore customary. Although single-reaction RT-PCR removes the need for a separate RT reaction, RNA isolation from clinical samples constitutes a major bottleneck in the diagnostic process, as it remains both manually laborious and expensive. Specifically, both the Charité University Hospital and the CDC protocols require the use of RNA purification kits, which not only results in a significant cost increase but led to a major supply shortage of such kits. It is therefore crucial that a new test is not only affordable, quick, and efficient, but also that it keeps the use of industrial kits to the minimum. Recent attempts have been made to circumvent RNA extraction in COVID-19 detection[7–9].

Here, we establish routines for SARS-CoV-2 RNA-extraction-free single-reaction RT-PCR testing (Fig. 1) on heat-inactivated nasopharyngeal swab samples in transport medium and compared the results with clinically diagnosed patient samples, demonstrating the viability of extraction-free SARS-CoV-2 diagnostics. In addition, we evaluate various buffer formulations as alternative transport media, and we provide data showing that SARS-CoV-2 RT-PCR can be performed in presence of high concentration of detergent, allowing testing directly on sample lysates. Importantly, our method builds on clinically established protocols and could easily be integrated to expand ongoing testing pipelines. It is also modular and can be incorporated into alternative approaches of detection utilising PCR.

## Results

**Development of SARS-CoV-2 hid-RT-PCR.** We started by investigating how transport media used for swab collection affect RT-PCR. To do this, we spiked synthetic full-genome SARS-CoV-2 RNA (SKU102024-MN908947.3, Twist Biosciences) into dilution series of three different transport media (Virocult MED-MW951S, Sigma; Transwab MW176S, Sigma, and Eswab 482 C, COPAN) used for clinical sampling at the time and place of the study (Karolinska University Hospital, Stockholm, Sweden). Each mock sample contained 50,000 synthetic SARS-CoV-2 gRNA copies and 95-0.1% medium, corresponding to 47.5–0.05% medium in the RT-PCR reaction. We performed single-reaction RT-PCR using 10 μl sample in a 20 μl reaction (TaqPath, Thermo, A15299) and the CDC nucleocapsid 1 (N1) primer-probe set (Table 1, Methods) and recorded cycle threshold ($C_T$) values for the dilution series of the media. We observed inhibitory effects in all three media and, importantly, pronounced variation between media (Fig. 2a). Virocult and Transwab demonstrated similar profiles of inhibition, resulting in +2-3 $C_T$ at the highest medium concentrations and minimal inhibition at concentrations below 30% medium in the RT-PCR reaction. Eswab completely impeded detection at high concentrations but reached a similarly low level of inhibition as Virocult and Transwab at 25% concentration in the reaction. To test that the batch of synthetic RNA used did not contain lingering DNA template, we additionally performed RT and qPCR reactions in two separate steps ("Methods") including RT ± controls, which demonstrated the lack of DNA amplification signal when the reverse transcriptase was excluded from the reaction (Supplementary Fig. 1a). Together, these results indicate minimal or no inhibition of Virocult, Transwab, and Eswab at

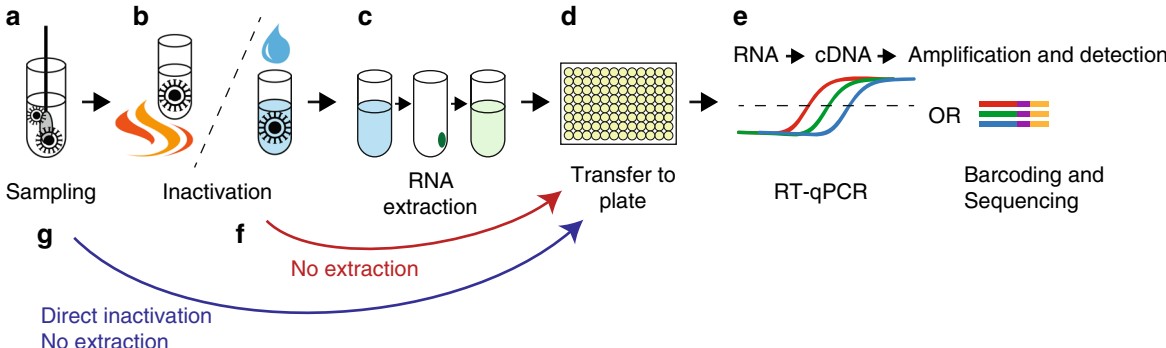

**Fig. 1 Schematic overview of SARS-CoV-2 RT-PCR testing procedure.** The currently widely used procedure for COVID-19 testing involves: **a** Collection of patient material and deposition of potential SARS-CoV-2 viral particles in transport medium. **b** Inactivation of the virus by detergent/chaotropic reagents or by heating. **c** RNA extraction. **d**, **e** Transfer to PCR-plate (96/384-well) format in which cDNA synthesis by RT and detection by qPCR may take place. Alternatively, detection can be made by sample barcoding and high-throughput DNA sequencing. **f**, **g** Unlike the widely used approach, which includes an RNA extraction step (c) using industrial RNA extraction kits, direct sample testing circumvents this process by omitting extraction. Instead, after clinical samples are deposited in transport medium, viral particles are inactivated either through heating or by direct lysis in detergent-containing buffer. The inactivated samples are then used for the downstream RT-PCR diagnostic reaction.

**Table 1 Primers and probes used for SARS-CoV-2 RT-PCR.**

| Name | Amplicon length (bp) | Description | Sequence (5′ to 3′) |
|---|---|---|---|
| N1 | 72 | Forward | GACCCCAAAATCAGCGAAAT |
| | | Reverse | TCTGGTTACTGCCAGTTGAATCTG |
| | | Probe | FAM- ACCCCGCATTACGTTTGGTGGACC -BHQ1 |
| E | 113 | Forward | GGAAGAGACAGGTACGTTAATA |
| | | Reverse | AGCAGTACGCACACAATCGAA |
| | | Probe | FAM- ACACTAGCCATCCTTACTGCGCTTCG -BHQ1 |
| RdRP | 81 | Forward | GTCATGTGTGGCGGTTCACT |
| | | Reverse | CAACACTATTAGCATAAGCAGTTGT |
| | | Probe | FAM- CAGGTGGAACCTCATCAGGAGATGC -BHQ1 |
| Rnase P | 65 | Forward | AGATTTGGACCTGCGAGCG |
| | | Reverse | GAGCGGCTGTCTCCACAAGT |
| | | Probe | FAM- TTCTGACCTGAAGGCTCTGCGCG -BHQ1 |

≤25% in the RT-PCR reaction, corresponding to ≤5 µl sample in a 20 µl SARS-CoV-2 RT-PCR reaction.

To test whether direct RT-PCR could accurately detect the presence of SARS-CoV-2 in clinical samples we started by obtaining five aliquots of nasopharyngeal swab samples stored in transport medium at −20 °C. Aliquots of the same samples had previously been clinically diagnosed using conventional RNA extraction (MagNA Pure 96 DNA and Viral NA SV Kit, Roche Diagnostics 06543588001) followed by RT-PCR, calling three patient samples as SARS-CoV-2 positive and two as negative to the virus (Clinical diagnostics performed at the Karolinska University Hospital, Stockholm). We inactivated the nasopharyngeal samples either by adding MagNA Pure 96 External Lysis Buffer (Roche, 06374913001), used in conventional RNA purification, or by heating at 65 °C for 30 min, and performed direct RT-PCR using 3 µl of sample. We observed a lack of amplification in SARS-CoV-2 positive samples inactivated with External Lysis Buffer (Fig. 2b, c and Supplementary Fig. 1b). However, RT-PCR performed directly on heat-inactivated samples correctly detected SARS-CoV-2 in all positive samples and lacked signal in the negative samples and controls (Fig. 2b, c). This indicated the viability in further exploring heat-inactivated direct RT-PCR (hid-RT-PCR) as a method to detect SARS-CoV-2 in clinical samples. We also tested two-step RT and qPCR for SARS-CoV-2 detection on the same clinical samples ("Methods"), correctly detecting the viral presence and absence (Supplementary Fig. 1c).

Next, we tested primer-probe set performance in hid-RT-PCR using nasopharyngeal swab samples and primers-probe sets targeting the SARS-CoV-2 genes RNA-dependent RNA polymerase (RdRP), envelope (E), and nucleocapsid (N1) (Table 1). We obtained additional heat-inactivated (65 °C 30 min) nasopharyngeal swab samples called as SARS-CoV-2 positive in previous clinical diagnostics ("Methods"). We observed, in our setting, a modest difference between N1 and RdRP in hid-RT-PCR (mean and median $C_T$ difference to N1: 0.63 and 0.27, $P = 0.032$, Wilcoxon signed-rank test) while the E-gene set appeared at considerably higher $C_T$ values than the other primer-probe sets (mean and median $C_T$ difference to N1: 2.9 and 1.7, $P = 0.00098$, Wilcoxon signed-rank test) (Fig. 2d, e), in line with previous results[10]. We argue that using short amplicon targets is critical for hid-RT-PCR due to the expected RNA fragmentation during heating, while considerations on the amplicon length should be less important for PCR amplification performed on extracted RNA from fresh samples. Due to the superior performance of N1 in hid-RT-PCR, we focused on this primer-probe set in the deepened analyses of hid-RT-PCR.

Our dilution experiments of medium and spike-in synthetic SARS-CoV-2 RNA had shown limited inhibition at ≤25% medium in the reaction (Fig. 2a). However, clinical samples contain additional material from the swab and other unknown and potentially inhibitory agents. In addition, due to potentially large variability between clinical samples, it is important to characterize inhibition curves in multiple individual clinical samples rather than an averaged mix of samples. The optimal amount of sample input in hid-RT-PCR should be a balance between possible inhibition from the sample and the amount of RNA going into the reaction. To identify the optimal range of sample input in clinical samples, we performed dilution series (10–0.01 µl) of individual COVID-19 positive heat-inactivated nasopharyngeal swab samples, and found an input of 1–4 µl sample in a 20 µl RT-PCR reaction to be optimal, avoiding the sharp inhibitory effect at higher amounts of sample input observed in some individual samples, yet minimizing $C_T$ (Fig. 2f, g).

To begin to explore whether direct RT-PCR on heat-inactivated samples might allow effective COVID-19 diagnostics, we performed heat inactivation (65 °C 30 min) of frozen (−20 °C) aliquots from 85 clinically diagnosed nasopharyngeal samples and performed hid-RT-PCR blindly to their COVID-19 status. We used 4 µl input and primers N1, RdRP, as well as RNase P for assessment of sample integrity. Thereafter, we combined the results of hid-RT-PCR with $C_T$ values from the clinical diagnostics performed on extracted RNA (MagNA Pure 96, Roche Diagnostics, 1:1 input-to-eluate volume, test targeting E and RdRP, Methods). We observed strong correlation between $C_T$ values of extracted and heat-inactivated samples (Fig. 2h) and overall agreement of SARS-CoV-2 calls (Fig. 2i). However, $C_T$ values for hid-RT-PCR (65 °C 30 min) on frozen samples were higher than for fresh RNA eluates of the same samples (median 6.7 $C_T$ difference) (Fig. 2h, i). This was expected given that (1) more RNA was loaded for eluates (2.5x, standard 10 µl input vs. 4 µl volume equivalent in hid-RT-PCR), (2) RNA extraction of eluates was performed on fresh samples while the aliquots used for hid-RT-PCR had been frozen and stored at −20 °C before heat inactivation, (3) heating may degrade RNA in presence of RNases and/or metal ions (metal-ion-based RNA cleavage). By performing RNA re-extraction from 19 freeze-thawed aliquots and comparing $C_T$ values to eluates of matched fresh aliquots of the same nasopharyngeal specimens we found the effect of freeze-thaw to result in +2-3 $C_T$ (Fig. 2j).

In summary, our data from 85 clinically diagnosed nasopharyngeal frozen samples showed that hid-RT-PCR could be a working option to extraction-based SARS-CoV-2 diagnostics, and that efforts to optimize the hid-RT-PCR protocol for maximum performance would be worthwhile.

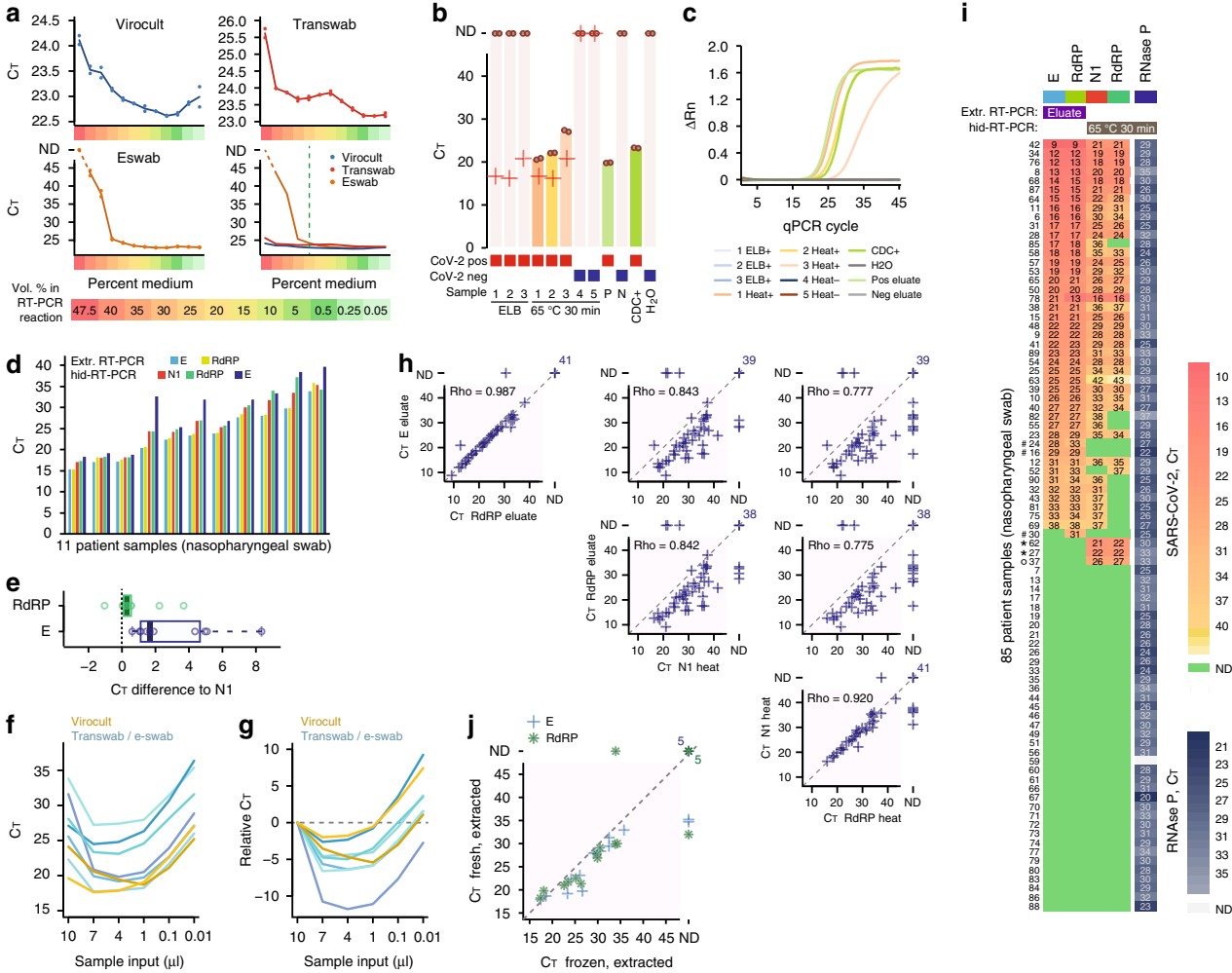

**Fig. 2 SARS-CoV-2 hid-RT-PCR on frozen nasopharyngeal swab samples. a** $C_T$ values from RT-qPCR performed on dilution series of transport medium (Virocult, Transwab, and Eswab) using 50,000 spiked copies of synthetic full-genome SARS-CoV-2 RNA and the N1 primer-probe set. Lines represent the mean of duplicates, shown individually as dots. ND: not detected. **b** Bar plots of $C_T$ from SARS-CoV-2 hid-RT-PCR on clinical nasopharyngeal swabs inactivated with MagNA Pure 96 External Lysis Buffer (ELB) or heat (65 °C 30 min). Dots indicate $C_T$ of hid-RT-PCR duplicates and crosses indicate $C_T$ values from diagnostics performed on fresh extracted RNA. Positive controls were extracted RNA from a positive sample (P) and a CDC positive control DNA plasmid (CDC+). Negative controls were extracted RNA from a negative sample (N) and water ($H_2O$). ND: not detected. **c** Amplification plots showing normalized reporter value ($\Delta Rn$, linear scale) as a function of qPCR cycle for the experiment and samples described in (**b**). **d** Bar plots of $C_T$ values of 11 positive nasopharyngeal swab samples using primer-probe sets targeting SARS-CoV-2 gene E, N, and RdRP. **e** Boxplots of $C_T$ difference in same samples as in (**d**) comparing E and RdRP with the N1 primer-probe-set. Center lines denote the median, hinges denote the interquartile range (IQR) and whiskers denote outlier points at maximum 1.5 × IQR. **f**, **g** Line charts of $C_T$ from individual clinical samples (colored lines) using variable amount of sample input. Shown as absolute $C_T$ (**f**) or $C_T$ relative to the 10 µl input (**g**). **h** Scatter plots of $C_T$ values from clinical diagnostics performed on extracted RNA (y-axis) and hid-RT-PCR (x-axis) of 85 nasopharyngeal swab samples, shown for different primer-probe set comparisons. Rho indicates Spearman correlation of positive samples. ND: not detected. **i** Heatmap of $C_T$ values from diagnostics performed on 85 clinical samples using extracted RNA (E, RdRP) and hid-RT-PCR (N1, RdRP), ranked by E gene $C_T$. Control for sample integrity by RT-PCR for RNase P in the same samples shown on the right. Two patients, marked with asterisk, were negative in extraction-based diagnostics but positive by hid-RT-PCR. The patients were later re-tested by extraction-based clinical diagnostics and confirmed to be SARS-CoV-2 positive. The patient marked with a ring was not re-tested. Three samples, marked with hash, were called COVID-19 positive by routine diagnostics but not by any primer-set in hid-RT-PCR. **j** Scatter plot of $C_T$ values from 19 matched fresh (y-axis) and freeze-thawed (x-axis) extracted samples, using the E gene (cross) and RdRP (star) primer-probe sets. ND: not detected. hid-RT-PCR shown in this figure was performed on previously diagnosed frozen samples.

**Optimisation of SARS-CoV-2 hid-RT-PCR.** To identify an optimal heat-inactivation program preceding hid-RT-PCR, we subjected 50 µl fresh (nonfrozen) aliquots of the same clinical nasopharyngeal samples in transport medium to different temperatures and incubation times (65 °C 30 min; 95 °C 5 min; 95 °C 10 min; 95 °C 15 min; and 98 °C 5 min; $n \geq 11$ patient samples). We observed consistent improvement (reduction) of hid-RT-PCR $C_T$ values in aliquots inactivated at 95 °C 5 min compared to 65 °C 30 min (median $C_T$ change: −1.3, $P = 1.1 \times 10^{-5}$, Wilcoxon signed-rank test, N1 primer-probe set) (Fig. 3a). Interestingly, all the other high-temperature (≥95 °C) conditions tested resulted in similar $C_T$ as the 95 °C 5 min treatment ($P > 0.05$, FDR-corrected Wilcoxon signed-rank tests) (Fig. 3a). Thus, we conclude that inactivation before hid-RT-PCR should be performed at 95–98 °C. This result is fortunate and important, since incubation at such high temperature should completely inactivate the virus[11]. Moreover, the stability across incubation times (5–15 min) at high temperature demonstrates a remarkable robustness of the procedure.

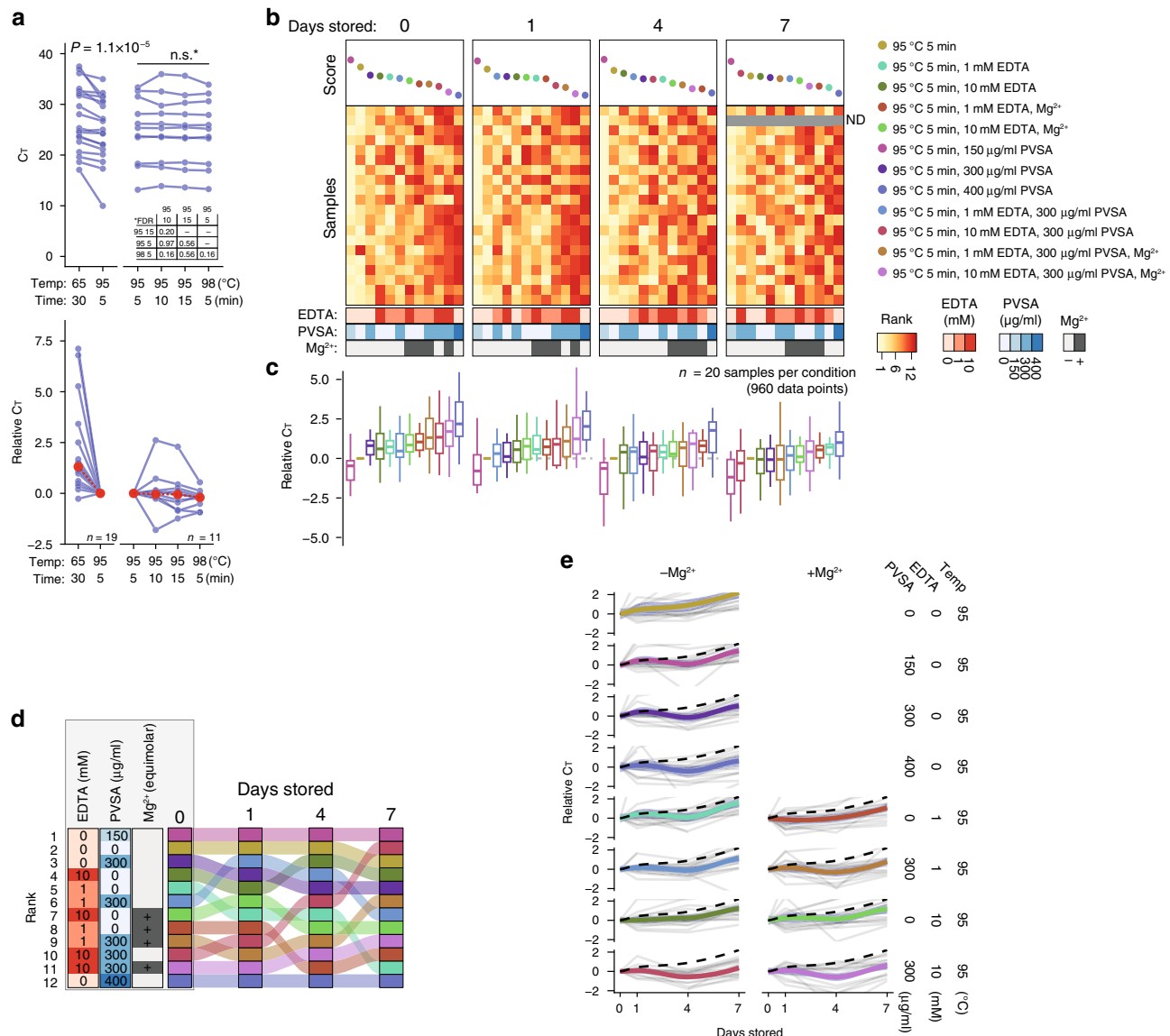

**Fig. 3 Optimisation of heat-inactivation conditions for SARS-CoV-2 hid-RT-PCR. a** $C_T$ values for aliquots of nasopharyngeal swab samples inactivated at different temperature and time conditions, shown on absolute scale (left) and $C_T$ change relative to the 95 °C 5 min condition (right). *P*-values calculated as two-tailed Wilcoxon signed rank tests. *FDR corrected for multiple testing. Median values shown in red. **b** Heatmaps of ranks based on absolute $C_T$ values for nasopharyngeal swab samples (*n* = 20) with or without addition of polyvinylsulfonic acid (PVSA) and/or ethylenediaminetetraacetic acid (EDTA) before performing heat inactivation at 95 °C for 5 min. For EDTA-containing samples, equimolar supplement of MgCl₂ was added to the RT-PCR reaction as indicated. The same samples were evaluated by RT-qPCR repeatedly at different number of days stored at 4 °C after the heat activation. All conditions are listed in Table 2. **c** Boxplots of $C_T$ values relative to the 95 °C condition without additives, ordered by the rank in (b). Center lines denote the median, hinges denote IQR and whiskers denote outlier points at maximum 1.5 × IQR. **d** Change in rank over days of storage at 4 °C. Colors and order same as in (**b**, **c**). Annotations refer to samples stored for 0 days. **e** Change in $C_T$ values over days stored at 4 °C across different conditions. Thick line and ribbon indicate fitted LOESS curve and ±95% confidence interval, respectively. The dashed line indicates the 95 °C (without additive) condition.

Next, we tested whether hid-RT-PCR could be improved by addition of the chemical, thermostable, RNase inhibitor poly-vinylsulfonic acid (PVSA), and/or the chelating agent ethylene-diaminetetraacetic acid (EDTA) during heat inactivation. PVSA halves in vitro RNase A activity (50% inhibition, IC₅₀) at the concentration 150 μg/ml and halves the RNase activity of E. coli lysate (IC₅₀) at 430 μg/ml[12]. At the same time, PVSA might inhibit RT-PCR. By performing dilution series of PVSA using synthetic SARS-CoV-2 RNA (SKU102024-MN908947.3, Twist Biosciences) as template, we identified a concentration range with prospective RNase inhibition in the sample yet limited inhibition of RT-PCR (Supplementary Fig. 1d). We then supplemented fresh aliquots of 20 COVID-19-diagnosed clinical nasopharyngeal

samples in transport medium with various amounts of PVSA and/or EDTA, and for EDTA-containing conditions we further performed tests supplementing equimolar amounts of MgCl₂ in the RT-PCR mix (12 conditions and 20 samples, *n* = 240) (Table 2). To additionally test the RNA stability in different treatments over time in storage (4 °C) we determined the SARS-CoV-2 RT-PCR $C_T$ change in the same samples over different number of days (0, 1, 4, and 7) after heat inactivation. We ranked the treatments within each sample and day, and we observed that 95 °C 5 min + 150 μg/ml PVSA produced the highest score (lowest $C_T$) followed by 95 °C 5 min without additives across day 0 to 4 (Fig. 3b–d). The benefit of EDTA and higher concentrations of PVSA only became apparent after 4–7 days in

**Table 2 Heat-inactivation conditions tested for SARS-CoV-2 hid-RT-PCR optimisation.**

| Condition | *EDTA (mM) | *PVSA (µg/ml) | #EDTA (mM) | #PVSA (µg/ml) | Mg$^{2+}$ |
|---|---|---|---|---|---|
| 95 °C 5 min | 0 | 0 | 0 | 0 | − |
| 95 °C 5 min, 1 mM EDTA | 1 | 0 | 0.15 | 0 | − |
| 95 °C 5 min, 10 mM EDTA | 10 | 0 | 1.5 | 0 | − |
| 95 °C 5 min, 1 mM EDTA, Mg$^{2+}$ | 1 | 0 | 0.15 | 0 | + |
| 95 °C 5 min, 10 mM EDTA, Mg$^{2+}$ | 10 | 0 | 1.5 | 0 | + |
| 95 °C 5 min, 150 µg/ml PVSA | 0 | 150 | 0 | 22.5 | − |
| 95 °C 5 min, 300 µg/ml PVSA | 0 | 300 | 0 | 45 | − |
| 95 °C 5 min, 400 µg/ml PVSA | 0 | 400 | 0 | 60 | − |
| 95 °C 5 min, 1 mM EDTA, 300 µg/ml PVSA | 1 | 300 | 0.15 | 45 | − |
| 95 °C 5 min, 10 mM EDTA, 300 µg/ml PVSA | 10 | 300 | 1.5 | 45 | − |
| 95 °C 5 min, 1 mM EDTA, 300 µg/ml PVSA, Mg$^{2+}$ | 1 | 300 | 0.15 | 45 | + |
| 95 °C 5 min, 10 mM EDTA, 300 µg/ml PVSA, Mg$^{2+}$ | 10 | 300 | 1.5 | 45 | + |

The concentration of ethylenediaminetetraacetic acid (EDTA) and polyvinylsulfonic acid (PVSA) in the samples during heat inactivation (star) and in the hid-RT-PCR reaction (hash) are indicated. The column labeled Mg$^{2+}$ indicates whether equimolar an amount of MgCl$_2$ was added to the hid-RT-PCR reaction to compensate for the chelating agent EDTA, which may sequester metal ions such as Mg$^{2+}$.

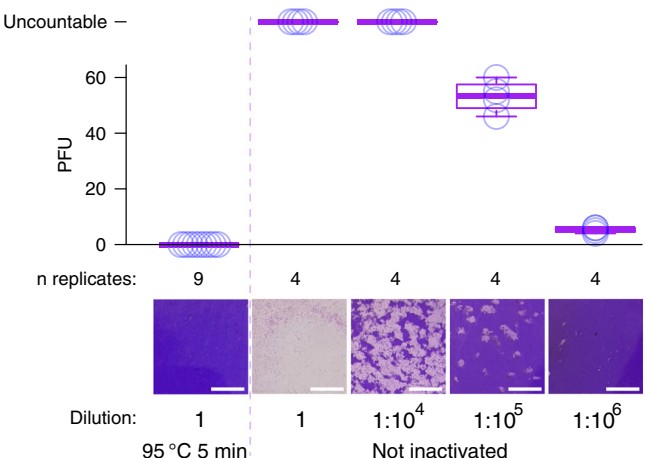

**Fig. 4 Heat-inactivation of SARS-CoV-2 confirmed by plaque assay.** Box plots showing the number of plaque forming units (PFU) observed after heat inactivation (95 °C 5 min, left) of in vitro propagated active SARS-CoV-2 as well as for a dilution series of SARS-CoV-2 without heat inactivation (right), with circles indicating the values of individual replicates. Center lines denote the median, hinges denote IQR and whiskers denote outlier points at maximum 1.5 × IQR. Virus was added to 9.6 cm$^2$ dishes seeded with 1 million Vero E6 cells and plaque assays were performed as described in "Methods". Undiluted samples contained ~2.5 million PFU of SARS-CoV-2. Representative images are shown below each condition (scale bar 5 mm).

storage (4 °C) (Fig. 3d, e). Interestingly, albeit C$_T$ values increased with time the results of the time series showed that heat-inactivated samples are surprisingly stable up to one week in storage at 4 °C.

Given the good performance of the 95 °C 5 min condition without additives (less than 1 C$_T$ difference compared 150 µg/ml PVSA, Fig. 3c) together with its simplicity in sample handling, we selected this condition for further benchmarking.

Finally, to experimentally validate the effectiveness of the thermal inactivation procedure, we propagated SARS-CoV-2 in vitro ("Methods"), and subjected active harvested viral particles (~2.5 million plaque forming units (PFU) in 500 µl) to heating at 95 °C for 5 min. We then performed plaque assays on Vero E6 cells, indeed demonstrating the lack of plaque formation from the heat-inactivated specimens (Fig. 4).

**Benchmarking of SARS-CoV-2 hid-RT-PCR.** Having optimised the heat-inactivation conditions, we next benchmarked SARS-CoV-2 hid-RT-PCR (95 °C 5 min) using the cobas 6800 system (Roche Diagnostics; hereby referred to as cobas) and a large set of paired clinical nasopharyngeal swab samples as reference. To first test the cobas performance compared to conventional RT-PCR on eluated RNA, we determined C$_T$ values of 21 purified clinical nasopharyngeal swab samples and performed a limit-of-detection experiment with the same sample (dilution from 1:100 to 1:100,000) on both systems, and observed a higher rate of detection and sensitivity for cobas (Supplementary Tables 1, 2). Given its performance and the fact that the cobas is a standar-dized and fully automated system (avoiding manual sample handling) we deemed it to be a suitable system for validation of SARS-CoV-2 hid-RT-PCR. We collected aliquots of 597 clinical nasopharyngeal swab samples diagnosed on the cobas analyser the same day using two targets (primer-probe sets towards ORF1 and E; "Target 1" and "Target 2" cobas SARS-CoV-2, P/N: 09175431190, Roche Diagnostics) and performed heat inactiva-tion (95 °C 5 min) followed by SARS-CoV-2 hid-RT-PCR (N1 primer-probe set, 4 µl sample input). C$_T$ values of hid-RT-PCR and cobas correlated well and had similar C$_T$ value distributions (P = 0.11 and 0.88; N1 vs. ORF1 and E, respectively; Kolmogorov–Smirnov test) (Fig. 5a, b). In addition to the 597 samples cross-compared with hid-RT-PCR, we plotted 9437 historical nasopharyngeal SARS-CoV-2 C$_T$ values collected on the same cobas machine and observed that the C$_T$ value distribution of the 597 samples was representative of the larger set (P = 0.23 and 0.35; ORF1 and E, respectively; Kolmogorov–Smirnov test) (Fig. 5c–d). Finally, we classified the COVID-19 status of the 597 nasopharyngeal swab samples on the cobas, either requiring positive signal (C$_T$ ≤ 40) for both targets (ORF1 and E), or, any target (ORF1 and/or E) to call a sample SARS-CoV-2 positive (Fig. 5e, leftmost bars). We plotted a heatmap of SARS-CoV-2 detection (C$_T$) and observed remarkable agreement between cobas and hid-RT-PCR (Fig. 5e–g). Using the diagnostic call of both cobas targets as reference, hid-RT-PCR had an accuracy, sensitivity, and specificity of 98.8% (95% confidence interval, CI$_{95}$: 97.5–99.5%), 96.0% and 99.8%, respectively (Table 3). Requiring only one cobas target to call samples SARS-CoV-2

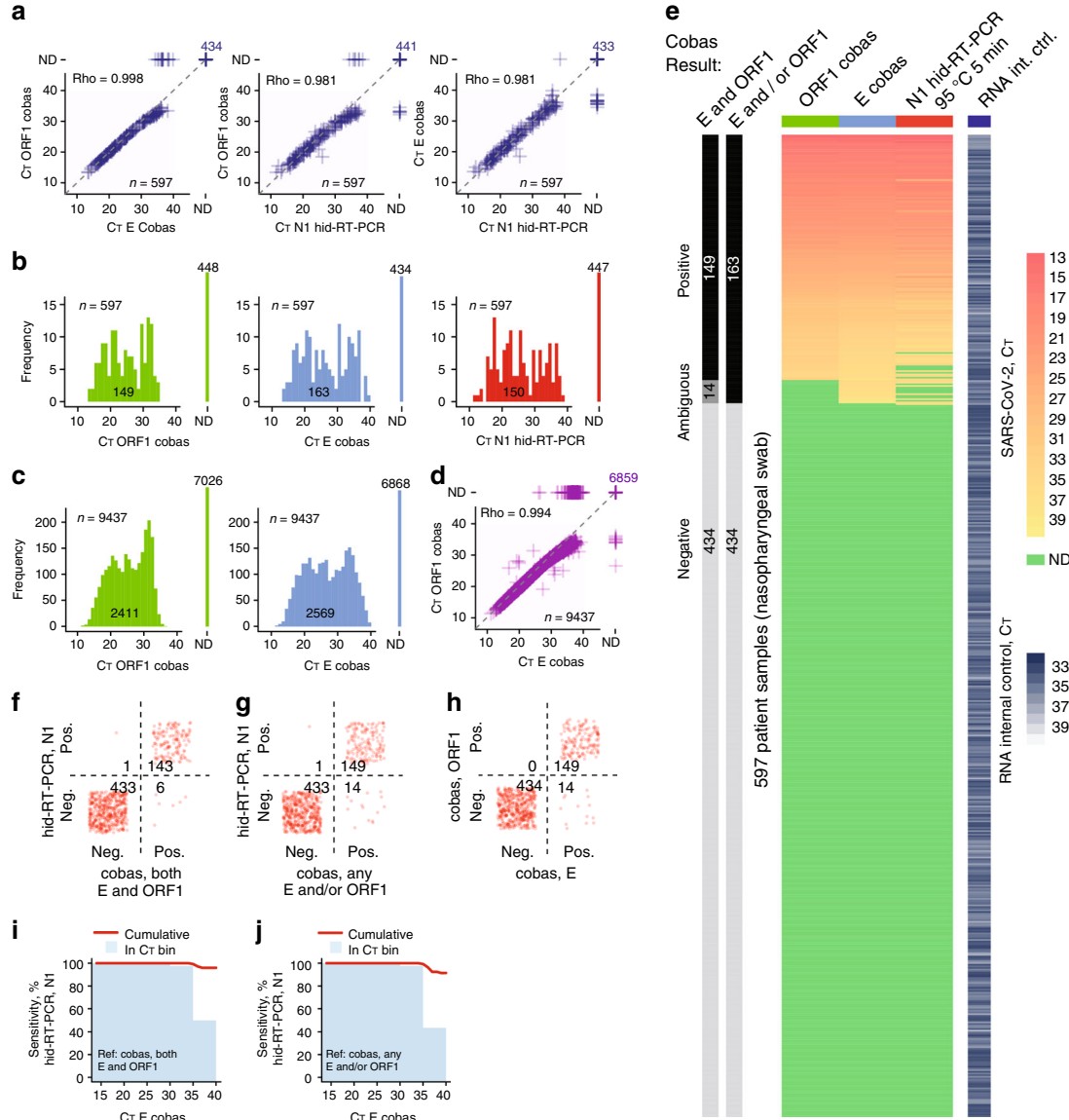

**Fig. 5 Validation of SARS-CoV-2 hid-RT-PCR. a** Scatter plots of $C_T$ values from clinical diagnostics performed on the Roche Diagnostics cobas 6800 (*y*-axis) and hid-RT-PCR (*x*-axis) for 597 nasopharyngeal swab samples, shown for different primer-probe set comparisons. Rho indicates Spearman correlation of positive samples. ND: not detected. **b** Histograms of $C_T$ values from 597 paired nasopharyngeal swab samples shown for the following primer-probe sets and conditions: cobas ORF1, cobas E, and hid-RT-PCR N1. ND: not detected. **c** Histograms of $C_T$ values from 9437 clinical samples analysed on a cobas 6800, show for ORF1 (left) and E (right). **d** Scatter plots of $C_T$ values from 9437 clinical samples analysed on a cobas 6800. Rho indicates Spearman correlation of positive samples. ND: not detected. **e** Heatmap of $C_T$ values from diagnostics performed on 597 clinical nasopharyngeal swab samples using the cobas 6800 (ORF1 and E primer-probe sets) and hid-RT-PCR (N1). The bars to the left indicate the clinical call from the cobas diagnostics, considering detection either in both or only one (any) primer-probe set for the diagnostic call. **f–h** Confusion matrix of diagnostic call from cobas 6000 and hid-RT-PCR (N1) for the data shown in (**e**). **i** Percent sensitivity of hid-RT-PCR (N1) using cobas ORF1 and E primer-probe sets as reference, shown as a function of $C_T$ threshold for the same sample as detected by the cobas primer-probe set E. The red lines denote the cumulative sensitivity below the given $C_T$ threshold and the bars denote the sensitivity in 5-$C_T$ bins. **j** Same as (**i**) but using cobas ORF1 and/or E primer-probe sets as reference for calculating sensitivity.

positive, the hid-RT-PCR accuracy, sensitivity, and specificity were 97.5% (CI₉₅: 95.9–98.6%), 91.4%, and 99.8%, respectively. We observed the performance of hid-RT-PCR (95 °C 5 min) using the N1 primer-probe set to be similar to the cobas ORF1 target (Table 3 and Fig. 5f-h). Next, we calculated the sensitivity of hid-RT-PCR at different $C_T$ thresholds and $C_T$ bins (as detected by the most sensitive cobas E-gene target; limit of detection ORF1: 0.009 TCID₅₀/ml, E: 0.003 TCID₅₀/ml; Supplementary Table 2) and found that hid-RT-PCR effectively only lost

in sensitivity for samples detected at $C_T$ 35–40 by the cobas E target, i.e., close to the limit of detection (Fig. 5i, j).

Together, these data demonstrate that a simple heat inactivation program followed by direct RT-PCR using the CDC primer-probe pair N1 detects SARS-CoV-2 with remarkable accuracy, sensitivity, and specificity given the ease of the method. As a result, RNA-purification-free SARS-CoV-2 detection is viable, enabling cheap, scalable, and rapid testing of COVID-19.

**Table 3 Accuracy, sensitivity and specificity of SARS-CoV-2 hid-RT-PCR.**

| Comparison | Accuracy (CI$_{95\%}$, $P$ value) | Sensitivity | Specificity |
|---|---|---|---|
| hid-RT-PCR (N1) vs. cobas, both primers (E and ORF1) | 98.8 (97.5–99.5, 4.8 × 10$^{-63}$) | 96.0 | 99.8 |
| hid-RT-PCR (N1) vs. cobas, any primer (E and/or ORF1) | 97.5 (95.9–98.6, 2.6 × 10$^{-60}$) | 91.4 | 99.8 |
| cobas ORF1 vs. cobas E | 97.7 (96.1–98.7, 1.8 × 10$^{-61}$) | 91.4 | 100.0 |

Accuracy, sensitivity and specificity of SARS-CoV-2 hid-RT-PCR (95 °C 5 min) using the Roche cobas 6800 analyzer as diagnostic reference. Each parameter was calculated using the cobas diagnostic call as reference, either requiring both cobas primer-probe sets producing signal ($C_T \leq 40$) to call a sample COVID-19 positive, or, requiring only one (any) cobas primer-probe set producing signal to call a sample positive. As a comparison, the parameters were calculated also for the cobas ORF1 primer-probe set, using the more sensitive cobas E primer-probe set as the reference (parameters calculated using the same samples as for hid-RT-PCR). Primer-probe sets used in the comparisons within brackets. Binominal test $P$ values. $N = 597$ fresh nasopharyngeal swab samples.

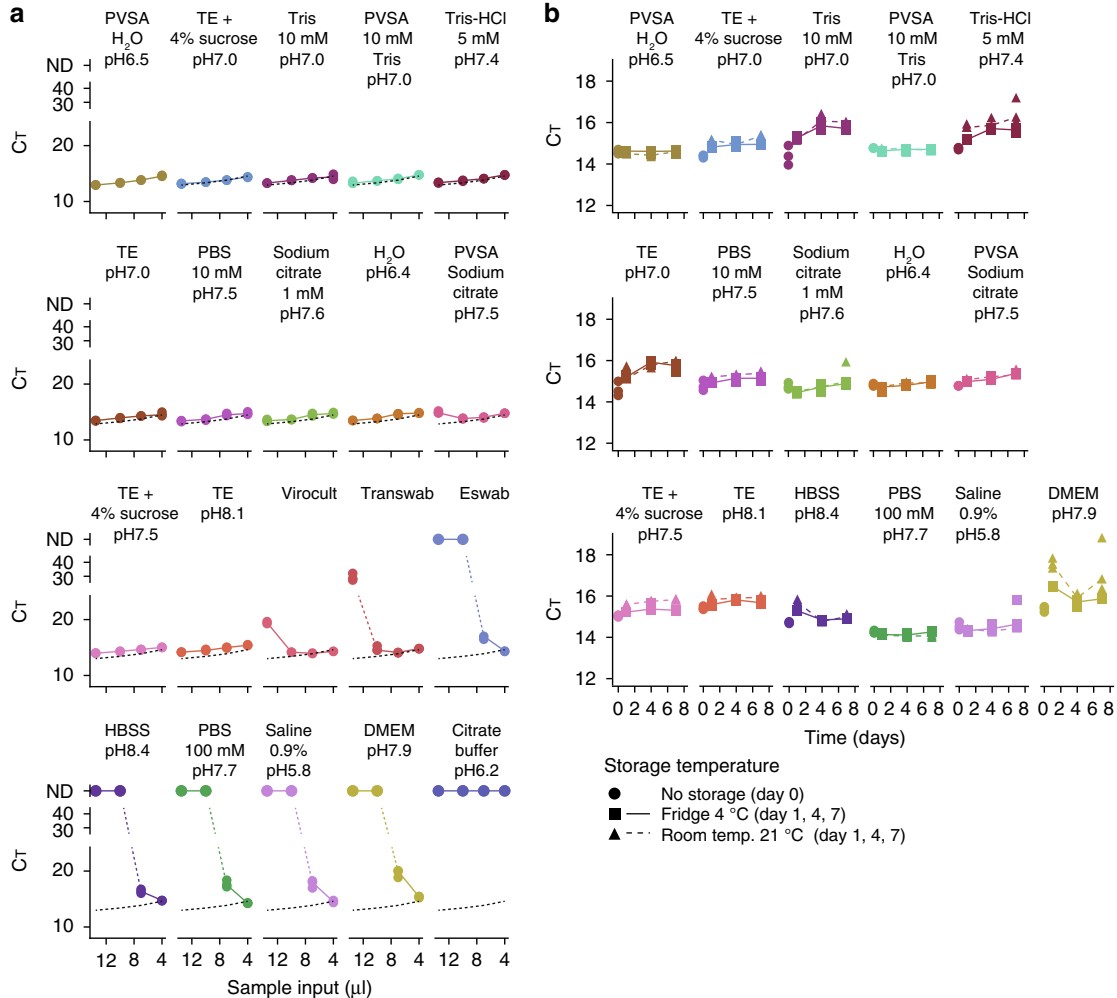

**Fig. 6 Identification of generic transport buffers optimal for SARS-CoV-2 hid-RT-PCR. a** Line charts of $C_T$ values (*y*-axis) from hid-RT-PCR (N1 primer-probe pair) for different volume of input sample (13.5, 10, 7, or 4 µl to a 20 µl reaction; *x*-axis), showing the inhibition profile of different transport buffers and media. The experiment was performed by adding equal amount of in vitro expanded SARS-CoV-2 to each buffer condition in experimental triplicates (dots). The dotted black lines indicate PVSA in H$_2$O condition, included for comparison. The buffers are ordered according to minimal $C_T$. **b** Line charts of $C_T$ values (*y*-axis) from hid-RT-PCR (N1 primer-probe pair, 4 µl input to 20 µl reactions) of in vitro expanded SARS-CoV-2 stored in different buffers for up to 7 days (*x*-axis) in fridge (4 °C, square) or room temperature (21 °C, triangle) before subjecting the samples to heat inactivation (95 °C 30 min) and hid-RT-PCR. The data are shown as individual replicates (points, $n = 3$) and median (line).

**Generic transport buffers optimal for direct RT-PCR testing.**
Our validation demonstrated the diagnostic potential of SARS-CoV-2 hid-RT-PCR for clinical nasopharyngeal samples collected in three commercial transport media (Virocult, Transwab, and Eswab). While media inhibition was successfully circumvented by dilution in the RT-PCR reaction (Fig. 2), this procedure reduced the input amount of sample (4 µl input per 20 µl reaction). Additional inhibition data for the widely used Universal

Transport Medium (Copan, not clinically used at the site and time of the study) is available in Supplementary Fig. 2. However, optimally for an extraction-free RT-PCR method, the swab material would be collected in a transport buffer that does not inhibit RT-PCR at all, since this would allow the input volume of sample to be maximized, improving the sensitivity of the assay. A simple and generic transport buffer formulation could also be a cheap alternative to commercial transport media and thus be

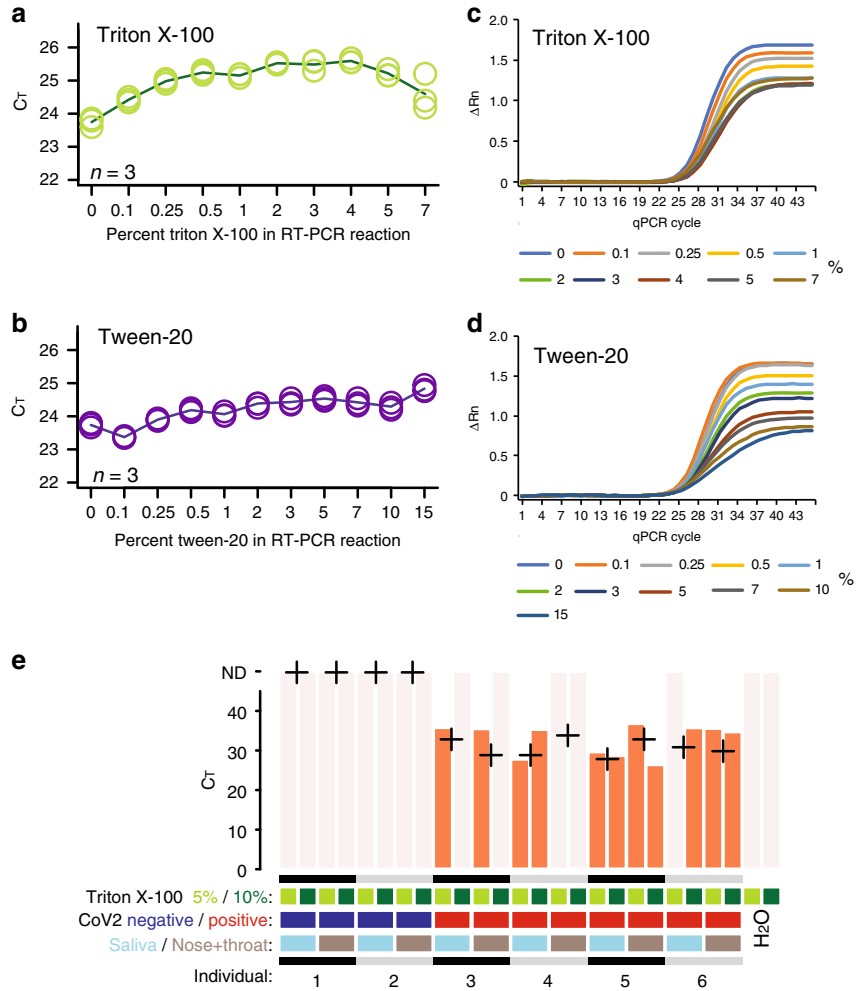

**Fig. 7 Direct SARS-CoV-2 RT-PCR detection from lysate. a**, **b** Line charts of $C_T$ values (y-axis) from RT-qPCR performed with different percent (vol./vol.) Triton X-100 (**a**) or Tween-20 (**b**) (x-axis) in the reaction. **c**, **d** Amplification plots showing normalized reporter value ($\Delta$Rn, linear scale) as a function of qPCR cycle for the experiments and samples described in (**a**, **b**). **e** Barplots of $C_T$ from SARS-CoV-2 RT-PCR using the N1 primer-probe set performed directly on lysed donor samples after storage and freeze thaw from self-sampling (saliva or nasal + throat swab suspensions taken with cotton tipped wooden sticks) without purification. Percent Triton X-100 indicates the percentage detergent in the sample (half concentration in the RT-PCR reaction). Crosses indicate $C_T$ values from diagnostics performed on extracted RNA from fresh aliquots.

suitable for mass testing. To identify such buffers, we spiked a fixed amount of active in vitro expanded SARS-CoV-2 into 17 different generic buffers (50,000 PFU into volumes of 100 µl performed in triplicates, "Methods") (buffers listed in Supplementary Table 3). Subsequently, we performed heat inactivation (95 °C 5 min) followed by RT-PCR (N1 primer-probe set) using 4, 7, 10, or 13.5 µl input to the 20 µl RT-PCR reaction to characterize inhibition. This identified several well-performing buffer formulations without inhibition at maximum input (13.5 µl per 20 µl RT-PCR reaction) such as for example PVSA in nuclease-free water (50 µg/ml), TE, Tris buffer and even nuclease-free water (Fig. 6a) (Note that DEPC-treated buffers are unsuitable, "Methods"). Interestingly, physiological saline solution (0.9% NaCl) and 100 mM PBS showed marked inhibition at higher input levels, with inhibition profiles similar to that of Eswab (Fig. 6a). Saline and 100 mM PBS are thus compatible with SARS-CoV-2 hid-RT-PCR at lower input volumes (4 µl input 20 µl RT-PCR reaction) but are clearly suboptimal choices of transport buffer for extraction-free RT-PCR.

A workable transport buffer must maintain detectable levels of SARS-CoV-2 several days after sample collection. We therefore also characterized how $C_T$ values changed with time in storage in the different buffers. We spiked active SARS-CoV-2 into the buffers (50,000 PFU in 100 µl) and stored aliquots in fridge (4 °C) and room temperature (21 °C) in triplicates for each condition. We collected aliquots after 1, 4, and 7 days in storage and subjected the samples to heat inactivation (95 °C 5 min) and RT-PCR (N1 primer-probe set), using 4 µl sample input in the 20 µl RT-PCR reaction for each buffer condition. We plotted $C_T$ values for each buffer, time-point and storage temperature and observed the $C_T$ change for the buffers (Fig. 6b). Interestingly, the chemical RNase inhibitor PVSA in water (pH 6.5) or 10 mM Tris buffer (pH 7) produced nearly unchanged $C_T$ throughout a week in storage in fridge as well as room temperature (Fig. 6b).

In summary, we have identified simple and affordable buffers suitable for SARS-CoV-2 hid-RT-PCR. Importantly, the buffers identified without RT-PCR inhibition allow more than threefold increase of sample input volume (13.5 vs. 4 µl), further improving the sensitivity of the SARS-CoV-2 hid-RT-PCR assay.

**Direct RT-PCR on lysed SARS-CoV-2 samples**. SARS coronavirus envelopes are self-assembled particles in which the lipid

bilayer is a weak spot[13], thus the viral envelope can be ruptured by surfactants and at the same time viral RNA can be released from similarly lysed human cells in the sample[14]. A direct route for SARS-CoV-2 screening could be self-testing using nose and throat swabs, or even on saliva, followed by lysis directly before RT-PCR on unpurified samples. RT-PCR assays directly on detergent-inactivated samples would require an RT-PCR assay resilient to high concentrations of detergent. We monitored the effect of Triton X-100 and Tween-20 on SARS-CoV-2 RT-PCR using spike-in of 50,000 copies of synthetic full-genome SARS-CoV-2 RNA (SKU102024-MN908947.3, Twist Biosciences) and the N1 primer-probe set. $C_T$ values were only modestly affected (+1–2 $C_T$) when incubated with as much as 5% Triton X-100 or 10% Tween-20 in the RT-PCR reaction (Fig. 7a, b). We observed lowered levels of fluorescence in the plateau phase in qPCR at increased concentrations of detergent, without markedly affecting the $C_T$ (Fig. 7c, d). To test whether actual SARS-CoV-2 RNA could be detected after direct lysis, we obtained six aliquots of saliva and six combined nose and throat swabs in PBS from six deidentified donors ("Methods"), of which four had been identified as COVID-19 positive and two as negative in extraction-based routine diagnostics (sample aliquots obtained from the Public Health Agency of Sweden). Notably, these samples were not collected by health care professionals using clinical grade flocked plastic swabs, rather the samples were self-collected using simple cotton swabs (deposition in PBS) and a jar without storage buffer for saliva (Methods). Furthermore, at the time of our experiment, these samples had been frozen, thawed and stored at room temperature for several hours combined. We tested these samples blindly, by mixing 5 µl sample (saliva or nose + throat swab) with 5 µl of 10 or 20% Triton X-100 and performed the SARS-CoV-2 RT-PCR using the N1 primer-probe set directly on these lysates (10 µl sample and Triton X-100 mix in 20 µl RT-PCR reactions, "Methods"). Indeed, all four COVID-19 positive donor individuals were correctly called as SARS-CoV-2 positive in at least one Triton X-100 condition or sample (saliva and/or nose + throat swab), while negative controls lacked signal (Fig. 7e).

These initial but principal results demonstrate that direct SARS-CoV-2 RT-PCR can also be applied on detergent-inactivated self-sampled material, opening an alternative route for large-scale population screening.

## Discussion

Scalable, rapid, and affordable COVID-19 diagnostics could help to limit the spread of SARS-CoV-2, consequently saving lives. However, RNA extraction constitutes a barrier to scale-up of testing. We explored procedures to circumvent RNA extraction by performing RT-PCR directly on heat-inactivated subject samples and sample lysates. Our results show that RT-PCR-based testing for SARS-CoV-2 infection can be performed through significantly simpler protocols and without the use of RNA extraction kits, nor other special kits. The results also show that this can be achieved without major sacrifice in accuracy of determining negative and positive cases. The procedure could be especially useful for massively scaling up SARS-CoV-2 testing, as the logistics and cost of RNA purification could be unworkable in mass testing. Importantly, the direct method is also attractive in settings where repeated, cheaper, and quicker testing is desirable, for example in frequent testing of healthcare personnel. The direct method that we present would also be compatible with downstream sequencing-based detection as an alternative to qPCR.

We determined RT-PCR inhibition profiles of different transport media as well as the optimal amount of reaction input of nasopharyngeal swab samples (Fig. 2). Further effort should be invested in similar characterization of many more brands and types of transport media in circulation. We propose that characterization of RT and PCR inhibition should become a standard (requirement) for commercial transport media in the future, as to assist direct testing in forthcoming epidemics. In the ongoing COVID-19 pandemic, we question whether it is at all suitable to deposit COVID-19 samples in conventional transport medium. Instead, we suggest that the swab material could be collected in a generic buffer that does not inhibit RT-PCR, especially as downstream viral culturing in lab is not meaningful for the vast majority of samples.

Heat inactivation cleaves RNA into shorter fragments, and thus primer and probe considerations in hid-RT-PCR should be more important for its sensitivity than for extraction-based RT-PCR of more intact RNA strands. Accordingly, we observed that the primer-probe set with the shortest amplicon (N1, 72 bp) performed best in hid-RT-PCR and the longest amplicon (E, 113 bp) performed the worst (Fig. 2d, e). Although we did not systematically discern whether relative abundance of SARS-CoV-2 gene copies might have contributed to the observed $C_T$ differences we underscore that short amplicon targets (such as N1) should be used in hid-RT-PCR due the expected RNA fragmentation.

After optimizing heat-inactivation conditions (Figs. 3 and 4), we validated hid-RT-PCR using the standardized and sensitive cobas 6800 system (Roche Diagnostics) as reference method (Fig. 5) and we observed high accuracy, sensitivity, and specificity of hid-RT-PCR (Table 3). We propose that the sensitivity of hid-RT-PCR could be improved even more if swab samples were to be collected in a buffer that do not inhibit RT-PCR, as the input amount of RNA could then be increased. Using controlled amounts of active SARS-CoV-2, we evaluated the performance of various buffer formulations in hid-RT-PCR, indeed identifying multiple buffers without RT-PCR inhibition (Fig. 6a) as well as capacity to preserve the SARS-CoV-2 signal over 7 days in storage in fridge as well as in room temperature (Fig. 6b).

Following this, we hypothesized that the direct RT-PCR pipeline for COVID-19 testing might also be implemented by sampling directly into a lysis buffer containing detergents such as Triton X-100. Lysed samples could immediately be subjected to RT-PCR analysis and diluted in the RT-PCR master mix without intermediate steps. Although the data is limited and the procedure should be refined, our results on self-collected saliva and nose/throat samples using cotton swabs (Fig. 7e) show that this strategy is workable. If the sensitivity ultimately proves to be adequate for meaningful decisions on self-isolation to limit spread, then this method could be applied to samples taken by the test-subjects themselves, allowing massive screening of the population.

## Methods

**Sample collection**. Clinical samples (nasopharyngeal swabs) were collected and deposited in transport medium (Virocult MED-MW951S, Sigma; Transwab MW176S, Sigma; or Eswab 482 C, COPAN) at the Karolinska University Hospital, Stockholm, Sweden. For routine diagnostics, 200 µl of sample in transport medium was inactivated by addition of 200 µl MagNA Pure 96 External Lysis Buffer (06374913001, Roche Diagnostics). Extraction was performed from 100 µl aliquots using MagNA Pure 96 DNA and Viral NA SV Kit (06543588001, Roche Diagnostics) or MGI Easy Magnetic Beads Virus DNA/RNA Extraction Kit (1000006989, MGI) and elution volume 100 µl. In this work we used anonymized or pseudo-anonymized surplus material from samples that had been collected for clinical diagnostics of SARS-CoV-2, in accordance with the Swedish Act concerning the Ethical Review of Research Involving Humans which allows development and improvement of diagnostic assays using patient samples which were collected to perform the testing in question. The samples used in direct lysis experiments (Fig. 7e) were deidentified self-collected volunteer samples from a COVID-19 screen performed in the Stockholm area, organized by the Public Health Agency of Sweden. For swab samples (nose + throat), two cotton tipped wooden swabs were used. Written instructs were provided to introduce one swabs into the throat via the chin as far back in the throat as possible and scrape for

10–20 s then rinse the swab in the provided buffer for 10–15 s. Instructions were also given to take a second wooden swab and introduce into the nose and scrape for 10–20 s in each nostril, followed by a 10–15 s rinse in the same buffer tube as the throat swab. The buffer in the swab test was 100 mM PBS pH 7.4. Further instructions were to leave a saliva sample at the same time by spitting 3–4 times in a small jar during a 5–10 min period. The samples were picked up and transported to a laboratory for testing typically within 1–10 h after the sampling. Samples were stored at +4 °C and RNA was extracted within 24 h and tested using RT-PCR. For the current study, samples were deidentified aliquots of these same samples that had been subsequently frozen at −20 °C for ~7 days, thawed and kept at room temperature for several hours, before performing the direct lysis experiments described herein. Informed consent was obtained from the participants.

**Ethics statement**. The intent of the work was clinical methods development as a response to the COVID-19 pandemic. The study was performed in accordance with the Swedish Act concerning the Ethical Review of Research Involving Humans. Ethical oversight and approval were obtained by the appropriate Swedish Authority (Dnr 2020-01945, Etikprövningsnämnden).

**Heat inactivation**. Nasopharyngeal swab samples in ~1 ml transfer medium were vortexed and 50 µl aliquots of each sample were transferred to 96-well PCR plates which were sealed (adhesive aluminum foil, VWR cat. 60941-112) and subjected to thermal inactivation using a thermal cycler with heated lid and using a thermal sealing mat. Alternatively, ~200 µl aliquots were inactivated in 1.5 ml Eppendorf tubes using a heat block. Before RT-PCR testing, the plates containing patient samples were centrifuged to collect debris in the bottom of the wells, and 4 µl sample for RT-PCR were collected from the liquid upper phase using a 10 µl multi pipette and added to plates containing 16 µl TaqPath mastermix. Each time the seal of a plate was opened we replaced the seal with a new seal to avoid cross contamination.

**One-step RT-PCR**. For reverse transcription and qPCR we used the one-step TaqPath RT-qPCR master mix (Thermo, A15299) according to manufacturer's instructions. Final reactions of 20 µl were formed by mixing 5 µl TaqPath master mix, primers-probe mix, sample, and RNase free water to fill the reaction. Primer and probe concentrations in the RT-PCR reaction were as follows, E and RdRP: 300 nM each primer, 200 nM probe; N1 and RdRP: 500 nM each primer, 125 nM probe. Primer and probe sequences are listed in Table 1. The thermal cycling steps were: 25 °C for 2 min, 50 °C for 15 min, 95 °C for 2 min, and 45 cycles of 95 °C for 3 s and 56 °C for 30 s. RT-qPCR was performed on a Step-One-Plus real time PCR machine (Applied Biosystems) using the StepOne Software v2.3. The samples from the self-test screen were subjected to the same protocol as described above but without heat inactivation. Briefly, samples (swab samples in PBS or pure saliva) were mixed with equal volume of 10% or 20% Triton X-100 at room temperature (~5 min) before performing RT-PCR using 10 µl sample-Triton X-100 mix as input to 20 µl RT-PCR reactions. Clinical COVID-19 diagnostics at the Karolinska University Hospital, Stockholm, Sweden were similarly performed using TaqPath and primer-probe sets for E and RdRP and 10 µl RNA eluate as sample.

**Two-step RT-PCR**. Reverse transcription was performed by mixing subject sample (4 µl in case of clinical nasopharyngeal swab sample), 1 µl 10 mM dNTPs, 0.15 µl 50 µM random hexamers (N8080127, Thermo), 0.1 µl RNase inhibitor (2313B, TaKaRa), 0.4 µl 0.5% Triton X-100 and RNase free water up to 4.5 µl, followed by incubation at 72 °C for 3 min. The sample was placed on ice and a mix containing 0.5 µl 100 mM DTT, 2 µl 5 M betaine, 0.1 µl 1 M MgCl$_2$, 0.25 µl RNase inhibitor (2313B, TaKaRa), 2 µl 5x Superscript II buffer, 0.5 µl Superscript II (Invitrogen) and water up to 5.5 µl was added. The samples were then incubated at 25 °C for 10 min followed by 42 °C for 25 min and finally for 70 °C for 15 min. Amplification of 10 µl cDNA (RT mix) using primers and probes described in Table 1 was performed using BioTaq DNA polymerase (Bio-21040, Bioline) in a 20 µl reaction containing 2 µl 10x NH4 Reaction Buffer (Bioline), 1.2 µl 50 mM MgCl$_2$, 0.2 µl 100 mM dNTP Mix and water up to 20 µl. The thermal cycling steps were: 25 °C for 2 min, 50 °C for 15 min, 95 °C for 2 min, and 45 cycles of 95 °C for 3 s and 56 °C for 30 s. qPCR was performed on a Step-One-Plus real time PCR machine (Applied Biosystems) using the StepOne Software v2.3.

**Synthetic full-genome SARS-CoV-2 RNA**. In control experiments with synthetic RNA (SKU102024-MN908947.3, Twist Biosciences), we used 50,000 copies per reaction. This arbitrary copy number was selected as to limit technical variation in the RT-PCR, and the copy number was calculated from the stock concentration provided by the manufacturer.

**Cobas 6800**. The cobas 6800 is a fully automated instrument that once samples have been loaded performs extraction, amplification, and detection. For details regarding the assay see Roche Diagnostics document 09179909001-01EN (Doc Rev. 1.0). Roche supplied reagents specific for the cobas SARS-CoV2 analysis are cobas SARS-CoV-2 (P/N: 09175431190) cobas SARS-CoV-2 Control Kit (P/N: 09175440190) cobas 6800/8800 Buffer Negative Control Kit (P/N: 07002238190).

Prior to analysis MagNA Pure 96 External Lysis Buffer (06374913001, Roche Diagnostics) were added to all samples at a ratio of 1:1 resulting in 150 ul of each patient samples being analysed. The instrument software version used was 01.03.08.1011.

**Gel electrophoresis**. Products of RT-PCR were separated by electrophoresis on a 3% agarose gel in 1x TBE buffer. The lengths of the products were determined relative to an Ultra-low Range DNA ladder (Thermo, SM1213). Images were taken using an Imagequant Las4000 camera system (Cytiva).

**SARS-CoV-2 in vitro assays**. SARS-CoV-2 (GenBank: MT093571.1), originally from the Public Health Agency of Sweden, was propagated on 90% confluent Vero E6 cells (ATCC-CRL-1586) for 3 days at 37 °C. Viral supernatant was separated via centrifugation at $300 \times g$ for 10 min and viral titers quantified by PFU. In brief, serial dilutions of viral supernatant were inoculated on 6-well plates seeded with 1 million Vero E6 cells per well (9.6 cm$^2$) for 1 h at 37 °C followed by removal of the inoculum media and two washes with PBS. Overlay medium consisting of 2:3 mix of 3% CMC and DMEM was added, and the plates were incubated at 37 °C for 3 days. Plates were then inactivated overnight at room temperature using 1 ml of 10% formaldehyde solution, washed and stained with crystal violet for 30 min, and plaques were counted manually. For confirmation of viral heat inactivation, 0.5 ml of the viral stock was pipetted into 2 ml safe-lock microcentrifuge tubes, which were incubated at 95 °C for 5 min, after which PFU were quantified as above. All work with active in vitro expanded SARS-CoV-2 was performed in the Biomedicum BSL-3 Core Facility at Karolinska Institutet.

**Tests of buffers for SARS-CoV-2 RT-PCR**. We obtained or prepared the buffers listed in Supplementary Table 3 and dispensed triplicates of 95 µl of each buffer into seven 96-well PCR plates. 5 µl viral stock, containing 50,000 PFU of in vitro expanded active SARS-CoV-2, was added to each well forming total volumes of 100 µl. Plates were subjected to heat inactivation (95 °C 5 min) in a thermocycler (LifeTouch 48-TC-96, BIOER) followed by hid-RT-PCR (using N1 primer-probe pairs) either directly after sampling (day 0) or after 1, 4, or 7 days in storage in fridge (4 °C) or room temperature (21 °C). At day 0 we performed RT-PCR using 4, 7, 10, or 13.5 µl sample input to a 20 µl one-step TaqPath reaction (data shown in Fig. 6a), and at day 1, 4, and 7 we performed RT-PCR using 4 µl sample input (data shown in Fig. 6b). Please note that DEPC-treated water and buffers were not used and are unsuitable because DEPC may inhibit PCR.

**Data analysis**. Graphical representations of data and statistical testing were performed using R. Effect of optimization parameters over time was modeled using linear regression using CT relative to day 0 as the response variable and time (days), EDTA, PVSA, and Mg$^{2+}$ additions as predictors both alone and as interaction terms with time. Accuracy, sensitivity and specificity was calculated using the confusionMatrix function in the caret R package[15].

**Reporting summary**. Further information on research design is available in the Nature Research Reporting Summary linked to this article.

## Data availability
Source data are provided with this paper.

## Code availability
Computational code is available at https://github.com/reiniuslab/COVID19.

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

### Acknowledgements

This research was supported and funded by the SciLifeLab/KAW national COVID-19 research program project grant (2020.0182), the Swedish Research Council (2017-01723) and the Ragnar Söderberg Foundation (M16/17) to BR. In accordance with the funding agencies' policy for COVID-19 related research, the results of this study were continuously made publicly available in preprint form (https://doi.org/10.1101/2020.04.17.20067348, first version online April 17, 2020). We are grateful to Sten Linnarsson for providing synthetic full-genome SARS-CoV-2 RNA, to Gerald McInerney for providing stocks of SARS-CoV-2, and to members of the Reinius lab for comments and discussions. In vitro work with active SARS-CoV-2 was performed at the Biomedicum BSL-3 Core Facility, Karolinska Institutet.

### Author contributions

I.S., M.E., N.R.S., N.P., J.o.A.a., H.S., and B.R. performed experiments. A.L. and B.R. analyzed data and prepared figures. M.E., M.V., S.M., A.G.R., J.A., B.H., and B.R. participated in sample handling and organization. BR conceived and supervised the study and wrote the manuscript. All authors participated in manuscript editing and approved the manuscript.

### Funding

### Competing interests

The authors declare no competing interests.
