## [Peer Review File · Nature Communications]

Reviewers' Comments:

Reviewer #3:

Remarks to the Author:

The article entitled "Massive and rapid COVID-19 testing is feasible by extraction-free SARS-CoV-2 RT-qPCR" by Smyrlaki et al, describes a series of logically planned and executed experiments to determine conditions for RT-PCR testing of nasopharyngeal samples for COVID-19 diagnosis. Authors first determined the relative level of inhibition that three sources of transport media have on the sensitivity of COVID-19 RT-PCR tests using known (500,000 viral RNA copies). The results indicated that there were minimal inhibition by two of the media and partial inhibition of the third medium, which disappeared with diluted media. In a second experiment, authors compared lysis buffer vs heat inactivation (65 oC/30 min incubation) for viral RNA extraction and determined that the sensitivity of both methods was equivalent in 85 clinical samples. Further experiments demonstrated that the benefits of using EDTA and PUSA are only apparent after 4 days of 4 oC storage. To further refine non-extraction methods and conditions, authors conducted a limit of detection study of 21 samples and compared conventional RNA extraction procedures vs a number of procedures and determined a 95 oC/5 min incubation produced comparative results to RNA extraction procedures. Moreover these 597 samples obtained CT values that compared with the historical data of 9437 previously obtained results. These findings enable cheaper, more scalable and rapid testing of samples, which is of high significance for the field of diagnostic testing of nasopharyngeal swabs, particularly in response to the current emergency, and given the limited availability of conventional RNA extraction procedures. The article is well written, and figures are conclusive. This reviewer's opinion is that these findings are of great interest to the laboratory community and will likely help increase testing capacity globally.

Critique (minor issues)

- 1- This method is not being run quantitatively; so I believe they should refer to the method as CoV-2 RT-PCR (not CoV-2 qRT-PCR)
- 2- Please succinctly say why the 500,000 RNA copies were chosen for the first experiment to determine inhibition of transfer media
- 3- Please briefly describe transfer media composition that may explain the differences in inhibition obtained.
- 4- The scales below Fig. 2a (% samples and % reaction) are not mentioned or described in the figure legend.
- 5- Line 77: change to: To test whether direct RT-qPCR could accurately detect the presence of SARS-CoV-2 in clinical...
- 6- Given that the results obtained with the hid-RT-PCR and the regularly extracted RT-PCR are similar; can the authors explain and defend the similarities or differences of the dilution factor in each method and how that may or may not explain the similarities in the results?

Reviewer #4:

Remarks to the Author:

Overall, this is a well-written paper with comprehensive presentation and interpretation of data. The design and conclusions are sound, and take into account all factors which are relevant to verifying the performance of the hid-RT-qPCR test. The idea is not novel as many have thought and tried this approach, but this is the first I am aware which documents and shows the development of the test, from various analytical tests, to testing with a good number of clinical specimens. The value of the paper will be of great interest to many countries would lack extraction reagents or who need faster, easier PCR tests. If this method can be packaged into a commercial kit that would accomplish its objectives even more. Below are some comments. Those marked * require more attention.

comments by line numbers:

36: the authors for Corman et al are from Germany, Netherlands, UK. I don't see China, France and Belgium.

37-39: optimized by US CDC ... through alternative kits... primer and probe sets. Does not make sense. It simply means developed different test protocol.

42-43: in the context of this article, good to emphasize that this first step of RNA purification involve both concentration and extraction steps, because direct PCR will skip the concentration step even if RNA is isolated

62: Should include commonly-used media specially for PCR like Universal Transport Medium (Copan) should be included. However, we understand the media chosen was based on institutional use.

94-95: use of comparative Ct values is at best a rough indication but not verification of relative performance, as it is the comparative LoD which is more important

99: could the superior performance of N be due more to relative abundance of N to start of with, rather than just the fragment length?

*117: you forget that loading 10 ul of eluate vs 4 ul direct sample does not include the concentrate of original sample before obtaining eluate. What was the starting volume used in the MagnaPure reaction?

The use of Cobas test as a comparator is good. The results analyzed by categorization and LoD are appropriate and within expectations. The conclusions of performance are sound.

*Where is a table comparing LoD between Cobas and direct hid-RT-qPCR?

It is unnecessary to compare Cobas to TaqPath, though having done so, that is good additional supporting data for using Cobas as a comparator.

REVIEWER COMMENTS

Reviewer #1

Didn't provide report

Reviewer #2:

Didn't provide report

Reviewer #3 (Remarks to the Author):

The article entitled "Massive and rapid COVID-19 testing is feasible by extraction-free SARS-CoV-2 RT-qPCR" by Smyrlaki et al, describes a series of logically planned and executed experiments to determined conditions for RT-PCR testing of nasopharyngeal samples for COVID-19 diagnosis. Authors first determined the relative level of inhibition that three sources of transport media have on the sensitivity of COVID-19 RT-PCR tests using known (500,000 viral RNA copies). The results indicated that there were minimal inhibition by two of the media and partial inhibition of the third medium, which disappeared with diluted media. In a second experiment, authors compared lysis buffer vs heat inactivation (65 oC/30 min incubation) for viral RNA extraction and determined that the sensitivity of both methods was equivalent in 85 clinical samples. Further experiments demonstrated that the benefits of using EDTA and PUSA are only apparent after 4 days of 4 oC storage. To further refine non-extraction methods and conditions, authors conducted a limit of detection study of 21 samples and compared conventional RNA extraction procedures vs a number of procedures and determined a 95 oC/5 min incubation produced comparative results to RNA extraction procedures. Moreover these 597 samples obtained CT values that compared with the historical data of 9437 previously obtained results. These findings enable cheaper, more scalable and rapid testing of samples, which is of high significance for the field of diagnostic testing of nasopharyngeal swabs, particularly in response to the current emergency, and given the limited availability of conventional RNA extraction procedures. The article is well written, and figures are conclusive. This reviewer's opinion is that these findings are of great interest to the laboratory community and will likely help increase testing capacity globally.

We thank the reviewer for the enthusiastic remarks, as well as for good suggestions how to further improve the study and manuscript. We certainly agree with the Reviewer that our study is of great interest and may help to increase the testing capability globally. To further strengthened the study, we have now included additional data using *in vitro* propagated SARS-CoV-2 showing that the heat-inactivation procedure that we advocate (95°C 5min) indeed efficiently inactivates the virus (new Fig. 4). Moreover, we have evaluated a large panel of generic buffer formulations as alternative to conventional transport and commercial viral transport media, identifying several generic buffers that are optimal for the direct SARS-CoV-2 RT-qPCR assay (new Fig. 6).

We have responded to each review comment point-by-point below. We hope that the reviewer will agree that our paper is now ready for publication, which may help to counter the ongoing COVID-19 pandemic.

Critique (minor issues)

1- This method is not beaning run quantitatively; so I believe they should refere to the method as CoV-2 RT-PCR (not CoV-2 qRT-PCR)

We agree with the reviewer that “RT-PCR” is the more appropriate term for the method and we have now updated the title, figures and manuscript text accordingly.

2- Please succinctly say why the 500,000 RNA copies were chosen for the first experiment to determine inhibition of transfer media

We used a copy number (50,000 copies) sufficiently large to limit technical variation in the RT-PCR experiment, but otherwise the choice of copies was arbitrary. We now mention the reason behind the chosen copy number in the Methods section. It reads: *“In control experiments with synthetic RNA (SKU102024-MN908947.3, Twist Biosciences), we used 50,000 copies per reaction. This arbitrary copy number was selected as to limit technical variation in the RT-PCR, and the copy number was calculated from the stock concentration provided by the manufacturer.”*

3- Please briefly describe transfer media composition that may explain the differences in inhibition obtained.

Commercial transport media evaluated have complex or undisclosed formulation and content concentration. Since there are numerous potential compounds in the commercial media that could inhibit RT-PCR we prefer not to speculate on what compounds contributed to the observed inhibition. Instead we hope that the reviewer will appreciate that we have now included additional experimental data (new Fig. 6) where we characterized the suitability of various generic buffer formulations for the direct SARS-CoV-2 RT-PCR assay. This new data is important since we indeed identified several suitable formulations without detectable inhibition to the RT-PCR. This information on alternative, simple and cheap sample buffers could be very valuable for implementing direct testing methods around the world.

4- The scales below Fig. 2a (% samples and % reaction) are not mentioned or described in the figure legend.

Thanks for noticing. We have now simplified the axis description in Fig. 2a and it should now be clear.

5- Line 77: change to: To test whether direct RT-qPCR could accurately detect the presence of SARS-CoV-2 in clinical...

Agreed and done.

6- Given that the results obtained with the hid-RT-PCR and the regularly extracted RT-PCR are similar; can the authors explain and defend the similarities or differences of the dilution factor in each method and how that may or may not explain the similarities in the results?

This is an important point. We indeed considered this in our experiments, but we realize that this did not come through in the manuscript text. The input-to-output volume in the extractions was 1:1 (*i.e.* without dilution or concentration, 100ul going in and 100ul going out; in accordance to the kit manufacturer’s instructions and the established practice in the clinical diagnostics unit at Karolinska Hospital, Stockholm). We now clearly stated this in the Results as well as in the Methods section.

Final remarks: We once again thank the reviewer for several good suggestions, all of which we have addressed, that helped us to improve the study and manuscript. We hope that the reviewer will now deem the manuscript suitable for publication.

Reviewer #4 (Remarks to the Author):

Overall, this is a well-written paper with comprehensive presentation and interpretation of data. The design and conclusions are sound, and take into account all factors which are relevant to verifying the performance of the hid-RT-qPCR test. The idea is not novel as many have thought and tried this approach, but this is the first I am aware which documents and shows the development of the test, from various analytical tests, to testing with a good number of clinical specimens. The value of the paper will be of great interest to many countries would lack extraction reagents or who need faster, easier PCR tests. If this method can be packaged into a commercial kit that would accomplish its objectively even more.

Thanks for these positive remarks. We are also thankful for the many good suggestions on how to improve the manuscript. We agree that our study and experimental data are important and of great interest where extraction-based tests are not feasible. Additionally, our method is highly suitable for frequent, repeated, testing. We are aware of other studies exploring extraction-free diagnostics (indeed our study was one of the early ones, available as preprint since mid April and submitted for review in early May), but as you suggest our study sticks out in terms of number of clinical specimens and analytical tests. To further strengthened our study, we have now included additional experiments making use of *in vitro* propagated active SARS-CoV-2. Importantly, we show that the heat-inactivation procedure advocated (95°C 5min) indeed efficiently inactivates the virus (**new Fig. 4**). Furthermore, we have now evaluated a large panel of generic buffer formulations as alternative to conventional transport and commercial viral transport media, identifying several buffers that are suitable for the direct RT-PCR-based assay (**new Fig. 6**). This additional information on alternative, simple and cheap sample buffers will be very valuable for implementing direct testing methods.

As you will see, we have responded to each of your comment point-by-point below. We sincerely hope that you will agree that our paper is now ready for publication without further delay. We believe that our data may help to counter the ongoing pandemic.

Below are some comments. Those marked * require more attention. comments by line numbers:

36: the authors for Corman et al are from Germany, Netherlands, UK. I don't see China, France and Belgium.

We double-checked the affiliation list for the authors of this paper and indeed found all mentioned countries listed. Please see the affiliation list pasted below:

1. Charité – Universitätsmedizin Berlin Institute of Virology, Berlin, Germany and German Centre for Infection Research (DZIF), Berlin, Germany
2. Tib-Molbiol, Berlin, Germany
3. GenExpress GmbH, Berlin, Germany*
4. Department of Viroscience, Erasmus MC, Rotterdam, the Netherlands
5. National Institute for Public Health and the Environment (RIVM), Bilthoven, the Netherlands
6. University of Hong Kong, Hong Kong, China
7. Université d Aix-Marseille, Marseille, France
8. Public Health England, London, United Kingdom
9. Department of Medical Microbiology, Vaccine and Infectious Diseases Institute, University of Antwerp, Antwerp, Belgium

However, since we do not know to what degree the stated affiliations reflect contribution to the protocol (the reviewer might know this better) we edited the sentence to the following: *“The currently used protocol was developed and optimized for the detection of the novel coronavirus at the*

Charité University Hospital, in collaboration with institutes in Germany, the Netherlands, China, France, United Kingdom and Belgium [4].”

37-39: optimized by US CDC ... through alternative kits... primer and probe sets. Does not make sense. It simply means developed different test protocol.

Agreed and corrected accordingly. The sentence now reads: “A different test protocol was developed by the Center for Disease Control (CDC) in the United States through comparison and validation of various kits for nucleic acid extraction and the use of alternative probe and primer sets for SARS-CoV-2 detection in clinical samples [5-6].”

42-43: in the context of this article, good to emphasize that this first step of RNA purification involve both concentration and extraction steps, because direct PCR will skip the concentration step even if RNA is isolated

We agree and have edited the mentioned sentence accordingly. It now reads: “Routinely, the application of quantitative RT-PCR (qPCR) for the relative quantification of a transcript of interest is preceded by (1) the isolation and purification of total RNA from the sample, (2) elution and possible concentration of the material, and (3) the use of purified RNA in a reverse-transcription (RT) reaction resulting in complementary DNA (cDNA) from the template RNA which is then utilized for the qPCR reaction.”

62: Should include commonly-used media specially for PCR like Universal Transport Medium (Copan) should be included. However, we understand the media chosen was based on institutional use.

*The reviewer correctly noted that the choice of medium tested was based on institutional clinical use (and product availability). As requested by the reviewer, we now included additional experimental data characterizing the inhibition profile of the Copan Universal Transport Medium (shown in **new Supplementary fig. 2**). Due to backorders at our supplier, it took us some time to get hold of this medium. In addition, we hope that the reviewer will appreciate that we have now also included additional experiments where we characterized suitability of various generic buffer formulations for the direct SARS-CoV-2 RT-qPCR assay (**new Fig. 6**), indeed identifying several formulations that do not have any detectable inhibition to the RT-PCR and preserve the virus signal in samples over time in storage. This information on alternative, simple and cheap sample buffers would be valuable for implementing direct testing methods.*

94-95: use of comparative Ct values is at best a rough indication but not verification of relative performance, as it is the comparative LoD which is more important

We again agree with the reviewer, and have therefore changed the wording of the mentioned passage accordingly: “We observed, in our setting, a modest difference between N1 and RdRP in hid-RT-qPCR (mean and median CT difference to N1: 0.63 and 0.27, $P=0.032$, Wilcoxon signed-rank test) while the E-gene set appeared at considerably higher CT values than the other primer-probe sets (mean and median CT difference to N1: 2.9 and 1.7, $P=0.00098$, Wilcoxon signed-rank test) (Fig. 2d-e), in line with previous results [10].”

99: could the superior performance of N be due more to relative abundance of N to start of with, rather than just the fragment length?

It is well established in experimental fields of RNA detection that amplicon length is highly important for detection in fragmented samples, such as heated specimens. Given the wide audience of the

paper we find it important to explicitly point out amplicon length consideration as one of the keys to success of the hid-RT-PCR protocol. Albeit not included in the present manuscript, we have indeed experimented with different amplicons for the very same gene (N gene) in heat-inactivated samples and observed higher rates of dropout of weak samples for the longer amplicons. Nonetheless, in the context of the included data we agree with the reviewer's reasoning in principal, and we have therefore highlighted the caveat in the Discussion section: *"Although we did not systematically discern whether relative abundance of SARS-CoV-2 gene copies might have contributed to the observed CT differences we underscore that short amplicon targets (such as N1) should be used in hid-RT-PCR due the expected RNA fragmentation."*

***117: you forget that loading 10 ul of eluate vs 4 ul direct sample does not include the concentrate of original sample before obtaining eluate. What was the starting volume used in the MagnaPure reaction?**

This is an important point – thanks for bringing it up. We indeed considered this in our experiments, but we realize that this did not come through in the manuscript text. The input-to-output volumes in the extractions were 1:1 (*i.e.* without dilution or concentration, 100ul sample going in and 100ul eluate going out; which is in accordance to kit instructions and the established practice in the clinical diagnostics unit at Karolinska Hospital, Stockholm, at the time of the study). We now stated this in the Results as well as the Methods section.

The use of Cobas test as a comparator is good. The results analyzed by categorization and LoD are appropriate and within expectations. The conclusions of performance are sound.

***Where is a table comparing LoD between Cobas and direct hid-RT-qPCR? It is unnecessary to compare Cobas to TaqPath, though having done so, that is good additional supporting data for using Cobas as a comparator.**

We agree that the cobas instrument is a good comparator, especially since the instrument has high sensitivity and throughput, and the sample handling is fully standardized and free of manual errors. The most important piece of the validation is the direct comparison between cobas and hid-RT-PCR using real clinical samples. Due to the high number of samples ($n=597$ in the benchmarking and $n=9437$ in the historically accumulated cobas data) we indeed spanned across the detection range of the cobas, as shown in Fig. 4b-c (**new Fig. 5b-c**). In the analysis, we further highlighted that samples close to the limit of detection in cobas (which are detected at \sim Ct 35-40 on the cobas instrument) had about 50% probability to result in dropout in hid-RT-PCR (see **Fig. 5b-d**). This is furthermore reflected in the experimentally determined sensitivity measure of hid-RT-PCR in **Table 3**. Altogether this provides a clear picture of the hid-RT-PCR assay's performance on relevant clinical material. Due to practical limitations we could not perform an artificial sample dilution series experiment for hid-RT-PCR with parallel cobas measurements of the same dilution series like we did for extraction-based RT-PCR in **Supplementary table 2**. This is simply due to policy reasons because the cobas instrument is currently fully dedicated to clinical diagnostics and not available for research purpose. The extraction-based RT-PCR vs. cobas dilution series data in **Supplementary table 2** on the other hand was generated during an evaluation phase of the cobas instrument. We hope that the reviewer agrees with us that the current benchmarking data together with all the additional aspects and strong points of the study stands without dilution series comparison between cobas and hid-RT-PCR.

Final remarks: We thank the reviewer for the many good comments which helped us to improve the study and manuscript. We sincerely hope that you will now deem the paper ready for publication.

Reviewers' Comments:

Reviewer #3:

Remarks to the Author:

Thanks for the through review of this manuscript and attention to all critiques. As a reviewer, I am satisfied with the corrections.

Reviewer #4:

Remarks to the Author:

all queries were answered well. this is an excellent paper. Recommend publication.

REVIEWERS' COMMENTS:

Reviewer #3 (Remarks to the Author):

Thanks for the through review of this manuscript and attention to all critiques. As a reviewer, I am satisfied with the corrections.

Reviewer #4 (Remarks to the Author):

all queries were answered well. this is an excellent paper. Recommend publication.

We thank the Editor and the reviewers for helping to improve the manuscript